# One Step Forwards in Knowledge of Blossom Blight Brown Rot Disease: *Monilinia* spp. SSR Marker Database

**DOI:** 10.3390/microorganisms12030605

**Published:** 2024-03-18

**Authors:** Raminta Antanynienė, Vidmantas Stanys, Birutė Frercks

**Affiliations:** Lithuanian Research Centre for Agriculture and Forestry, Institute of Horticulture, Department of Orchard Plant Genetics and Biotechnology, Kaunas District, LT-54333 Babtai, Lithuania; raminta.antanyniene@lammc.lt (R.A.); birute.frercks@lammc.lt (B.F.)

**Keywords:** *Monilinia* spp., database, microsatellite, molecular markers

## Abstract

A freely available *Monilinia* spp. marker database was created, containing microsatellite (SSR) data of the three most essential European fungal pathogens: *M. fructigena*, *M. laxa*, and *M. fructicola*. These pathogens cause brown rot blossom blight. Microsatellites were identified using the bioinformatics tool Genome-wide Microsatellite Analyzing Toward Application (GMATA). The database provides information about SSR markers: forward and reverse sequences of the primers, fragment sizes, SSR motifs (and repeats), and the exact locations with the coordinates in the reference genome. This database currently contains information about 39,216 SSR motifs and 26,366 markers. In total, eight primers generated in silico were validated experimentally and they are marked in the database. All scientists can join this collaboration by adding their experimental data. This database is the initial start of organizing *Monilinia* spp. molecular data worldwide and, in the future, it could be extended by adding more molecular and genomic information.

## 1. Introduction

The significant losses of fruit yield during the plant vegetation period and postharvest due to reduced shelf-life are caused by fungal *Monilinia* spp. pathogens [1,2]. The three most essential and common *Monilinia* species in Europe are*: M. fructigena* Honey, *M. laxa* (Aderh ir Ruhland) Honey, and *Monilinia fructicola* (G. Winter) Honey [3,4]. The pathogens are spread all around the world, and their distribution depends on the geographical location and the host plant [2]. For the control of *Monilinia* spp., it is crucial to track this disease’s epidemics and genetic variations [5].

DNA markers are used for reliable pathogen genetic analysis. The genetic diversity of *Monilinia* spp. has been widely studied with polymerase chain reaction (PCR) and molecular markers like inter simple sequence repeat (ISSR), random amplified polymorphic DNA (RAPD), restriction fragment length polymorphism (RFLP), amplified fragment length polymorphism (AFLP), and simple sequence repeats (SSRs)/microsatellites [5,6,7,8,9,10,11,12].

Microsatellites are preferred molecular markers because they are highly polymorphic, easy to use, highly versatile, and cheaper than other methods [13]. SSRs are short tandem repeats of DNA motifs and can be found in the whole genome—in protein-coding and non-coding regions. Due to their high variability, microsatellite sequences are crucial for genome evolution [14,15].

For *Monilinia* spp. pathogens, considerable effort has been made to analyze and develop SSRs. The first reports of SSR motif research of *Monilinia* spp. pathogens were published in 2003 when specific nested primer pairs were developed for the *M. fructicola* pathogen [16]. To date, 20 SSR markers have been developed for *M. fructicola* [12] and only one for *M. laxa* [5]. Therefore, there is a demand for more microsatellite markers for analysis of *Monilinia* spp. pathogens’ genetic diversity and pathogenicity.

Conventionally, SSR marker development is based on screening genomic libraries. However, this method is time consuming, requiring priori information about the DNA sequences, and expensive [17,18]. A more rapid and cost-effective way is to use genomic sequences, which are available in genome databases such as the National Centre for Biotechnology Information (NCBI) [19]. To date, improved bioinformatics tools are used for efficient SSR identification, characterization, and SSR marker development in silico [13].

The genetic data available in the database allow for the analysis of pathogens at the genomic and transcriptomic levels [20]. Since 2017, 15 genomes of six *Monilinia* species have been announced and are available in the NCBI database [19]. All genomes are announced at the contig or scaffold level, and only *Monilinia vaccinii-corymbosi*’s genome is announced at the chromosome level (GCA_017357885.1) [19]. However, the genomes of other *Monilinia* spp. exist only at the draft-reference genome level [21,22,23]. The transcriptomes of the most common *Monilinia* species, namely *M. fructigena*, *M. fructicola*, and *M. laxa*, have been analyzed [20]. All three transcriptomes were similar in functional categories by their unigene distribution profiles. The transcripts are involved in the fungi’s diversity, development, and pathogenicity with the host plant [20].

The ever-growing volume of genetic and genomic information raises the need to organize molecular data as structured information in separate databases and to make it accessible to researchers worldwide. Therefore, the aim of this study was to analyze the structure of SSR markers distributed in the genomes of the three most common *Monilinia* spp. pathogens worldwide, creating a publicly available molecular marker database that allows researchers to identify, compare, and assess molecular data for specific needs.

## 2. Materials and Methods

For the development of microsatellite primers in silico, the genome assemblies of *M. fructicola* (ASM869222v1) [22], *M. fructigena* (ASM367162v1) [23], and *M. laxa* (ASM929945v1) [21] were used from the NCBI genome database [19].

Microsatellite motifs for *M. fructicola*, *M. fructigena*, and *M. laxa* were identified separately using Genome-wide Microsatellite Analyzing Toward Application (GMATA) v2.2_Build_20180415 software [13]. For microsatellite identification, the GMATA module “SSR identification” was used with parameters set as follows: the minimum length of the motifs was 2, the maximum length was 6, and the minimum number of repetitions of the motifs was 2. The GMATA module “SSR statistical plotting” was used to plot graphs of the top k-mers and to analyze the top distribution of SSR length and grouped motifs distribution.

Genome-wide SSR primers were designed using the Primer3 algorithm integrated within GMATA software’s “marker design” module with default parameters. The suitability and polymorphism of the identified microsatellite motifs for their use as microsatellite markers were verified using the “e-mapping” module of GMATA software with default parameters.

For experimental SSR primer validation, isolates of *Monilinia* spp. were collected in the orchard of the Institute of Horticulture at the Lithuanian Center for Agriculture and Forestry in the summers (1–2 weeks of August) of the year 2020 and 2021. Fruits of plum and apple with visible brown rot symptoms were collected randomly. In total, 88 isolates (62 from plum, 26 from apple) were collected. The brown rot mycelium was collected into 2 mL microcentrifuge tubes. The pathogen cultures were isolated and grown on potato dextrose agar (PDA) (Scharlab, Barcelona, Spain) according to the method in [24]. The cultures were incubated for 10 days in an incubator (Memmert, Schwabach, Germany) at 22 °C in 16/8 h day/night mode.

Total fungal genomic DNA was extracted from a 10-day-old culture using the Genomic DNA Purification Kit (Thermo Scientific, Waltham, MA, USA) according to the manufacturer’s instructions. The quality and concentration of extracted DNA were evaluated using a NanoPhotometer™ (Implen, München, Germany) and visualized in 1.5% agarose gel under UV light using “E.A.S.Y Win 32” (Herolab, Wiesloch, Germany). Samples were stored at −20 °C until further analysis.

*Monilinia* spp. were identified by multiplex PCR using four published primers: MO368-8R—specific for *M. fructigena* (402 bp) and *M. polystroma* (425 bp), MO368-10R—specific for *M. fructicola* (535 bp), Laxa-R2—specific for *M. laxa* (351 bp), and MO368-5—reverse primer, common for all *Monilinia* spp. [6]. Multiplex PCR reactions were performed with a 20 μL total volume of the reaction mixture, consisting of 2 μL of 10 × PCR, 2 μL of dNTP Mix, 0.7 μL of each primer, 0.1 μL of Taq DNA polymerase (Thermo Scientific, USA), 1 μL of extracted DNA template, and 8.5 μL nuclease-free water. The thermocycling conditions consisted of initial denaturation and polymerase activation at 94 °C for 10 min; then 7 cycles of 94 °C for 30 s, 59 to 65 °C for 45 s, and 72 °C for 60 s; followed by 25 cycles of 94 °C for 30 s, 58 °C for 45 s, and 72 °C for 60 s; and a final elongation process at 72 °C for 10 min. For the annealing stage, a gradient (−1 °C) was chosen according to the original temperatures described in the manufacturer’s instructions for each primer pair. Electrophoresis was carried out, and the PCR products were observed in 1.5% agarose gel using a GeneRuler 100 bp DNA Ladder (Thermo Scientific, Waltham, MA, USA) as a marker. The results were visualized under UV light using “E.A.S.Y Win 32” (Herolab, Wiesloch, Germany).

To evaluate the polymorphism of *Monilinia* spp., microsatellite analysis was performed. To identify microsatellites in DNA sequences associated with pathogenesis, 11 hypothetically pathogenesis-related proteins of *M. fructicola* [20] were selected, and their DNA sequences were compared with the genomes of *M. laxa* and *M. fructigena*. The overlapping DNA sequences in all three genomes of *Monilinia* spp. were used for further analysis. A multiplex FASTA file was generated from the selected DNA sequences, and SSR marker research was carried out using the GMATA platform. In silico PCR (ePCR) was performed via the “e-mapping” module in the GMATA program to test if newly designed SSR markers were species-specific.

Experimental validation of the created SSR primers under laboratory conditions was performed for eight primer pairs specific for the *Monilinia* spp. pathogens (Table 1). Isolates of *Monilinia* spp. were amplified using the mixture described above, with reaction mixtures prepared for each created primer. The thermocycling conditions were as described above. The results were observed after gel electrophoresis in 1.5% agarose gel under UV light. According to the specificity of the primers obtained experimentally, four primers were selected and labelled with a blue fluorescent dye—FAM. PCR amplifications were performed with a 10 µL total volume of the reaction mixture, consisting of 300 ng/µL DNA, 0.2 mM of each primer, 500 U Taq DNA polymerase, 2 mM dNTP, 25 mM MgCl, 10× buffer, 1% PVP, and 10 mM DTT (Thermo Scientific, Waltham, MA, USA). Experiments were carried out three times on two biological replicates. Only reproducible data were evaluated.

Amplification of DNA fragments was performed in a thermocycler (Eppendorf, Hamburg, Germany) as described above, with seven cycles and a touchdown procedure at the primer annealing step, comprising 30 s at 94 °C, 45 s at 62 °C (−1 °C in each cycle by touchdown procedure), and 1 min at 72 °C, followed by 25 cycles of 30 s at 94 °C, 45 s at 55 °C, and 1 min at 72 °C. The results were observed by capillary electrophoresis using an ABI 3130 (Applied Biosystems, Foster City, CA, USA) gene analyzer, using the GeneScan 500LIZ standard (Thermo Scientific, Waltham, MA, USA).

The primary data were analyzed using the GeneMaper 4.0 program (Applied Biosystems, Foster City, CA, USA) and extracted to a Microsoft Excel file. Expected (*H_e_*) and observed (*H_o_*) heterozygosity and polymorphism information content (*PIC*) were calculated using the PowerMaker program [25]. The informativeness of microsatellite primer pairs was assessed according to the method in [26].

The SSR marker pool was created according to the data generated with the GMATA program. Output files (“.ssr” and “.ssr.mk”) were adjusted by correcting marker names and content was prepared for uploading to the database using the Notepad++ v7.9.1 program). The *Monilinia* spp. marker database was constructed using Linux Server as the computer operating system. The SSR data were processed using R programming language [27]. The web application was created using the Shiny framework [28]. The database’s content was managed using the MongoDB 6.0 program [29].

## 3. Results

In total, 39,216 microsatellite motifs were identified in the genomes of three *Monilinia* spp. (Table 2). The highest number of SSR motifs was identified in the *M. fructicola* genome (15,788 motifs), followed by the *M. laxa* genome (12,337), and the lowest number of SSRs was identified in *M. fructigena* (11,091). The biggest genome of all *Monilinia* spp. was that of the *M. fructicola* pathogen (44,048 Mbp), and the density of SSR motifs was the highest (359 per Mbp).

In total, 26,366 SSR markers were developed for *Monilinia* spp.: 9754—*M. fructicola*, 8506—*M. fructigena*, and 8106—*M. laxa* (Table 2). Species-specific SSR markers were identified by in silico ePCA: 98.6% markers were specific for the *M. fructicola* genome, 96.3% —*M. fructigena*, and 96.0%—*M. laxa*.

The distribution of the microsatellite motifs in the *Monilinia* spp. genomes was non-uniform. In the *M. fructigena* genome, 55.5% of all SSR motifs were found in the first scaffold of the genome. In the *M. fructicola* and *M. laxa* genomes, SSR motifs were distributed evenly over all scaffolds (Figure 1A). The length of SSR motifs in the *Monilinia* spp. genomes ranged from 10 to 36 bp. The most frequent SSR motifs in *Monilinia* species were 10 bp in length: *M. fructicola*—24.6%, *M. fructigena*—19.7%, and *M. laxa*—26.2% (Figure 1B).

The structure of *Monilinia* spp. SSR motifs was between 2 and 6 k-mers (Figure 2). The highest number of SSR motifs was dimers (45.3%), followed by trimers (19.5%), tetramers (18.7%), pentamers (8.8%), and hexamers (7.7%) for all three *Monilinia* species. The highest number of dimeric structure SSR motifs was found in the *M. fructicola* pathogen (7178), followed by the *M. laxa* (6036) and *M. fructigena* (4590) genomes. Comparing all *Monilinia* spp., the lowest numbers of trimer (2113) and tetramer (1814) structure motifs were observed in the *M. laxa* genome; however, the lowest numbers of pentamers (1000) and hexamers (868) were in the *M. fructigena* genome. The most frequent dimers were ‘TA’ and ‘AT’ in the *M. fructicola* genome, with 15% and 13.6%, respectively; in the *M. fructigena* genome—11.2% and 10.9%, respectively; and in the *M. laxa* genome—18.2% and 15.1%, respectively.

DNA sequences of 16 hypothetical pathogenesis-related proteins in the *M. fructicola* genome were identified according to [20] data. After DNA sequence analysis with the NCBI Blastn 2.13.0 program, 11 hypothetical pathogenesis-related (PR) proteins observed in all three *Monilinia* spp. were selected for further analysis. In the DNR sequences of these PR proteins, a total of 2745 SSR motifs were identified for *Monilinia* spp. and 470 species-specific primers were generated (113—*M. fructigena*, 188—*M. fructicola*, 169—*M. laxa*) (Appendix A, Table A1).

The highest number (33%) of microsatellite motifs was observed in EYC_2169 PR protein DNA sequences in *M. fructicola* and *M. laxa* species. In *M. fructigena*, the highest number of SSR motifs were observed in EYC_8759 PR. The amount of SSRs in the protein DNA directly depended on the length of the protein sequence—the longer the sequence, the more microsatellite motifs were detected. Dimers were predominant in all PRs of *Monilinia* spp., with an average of 69.6% of all sequences, followed by trimers (26.5%) and tetramers (3.9%). The most frequent dimer motif in all *Monilinia* spp. sequences was ‘AT’, it was observed in 11.3% of all SSRs in *M. fructigena*, *M. fructicola*—10.9%, and *M. laxa*—9.9%. The second most frequent motif in the *M. fructigena* (8%) and *M. laxa* (9.4%) pathogens was ‘TC’; however, in the *M. fructicola* pathogen’s sequences, the ‘GA’ (8.5%) motif was observed more often than ‘TC’ (8.4%).

The specificity of the developed primers was evaluated under laboratory conditions for the *Monilinia laxa* and *Monilinia fructigena* pathogens. The fragments of primers ML104 (358 bp), MFg90 (292 bp), and MFg39 (346 bp) were observed for both pathogens of *Monilinia* spp. The results were controversial to those observed under in silico conditions. Three primers, ML159 (313 bp)—specific for the *M. laxa* pathogen, and MFg27 (239 bp) and MFg2 (260 bp)—specific for *M. fructigena* under in silico conditions, were amplified in both species.

The SSR primers ML2 (348 bp) and ML86 (336 bp) were specific for the *M. laxa* pathogen. ML2 primer (348 bp) did not amplify any fragments in the *M. fructigena* samples (Figure 3). ML2 primer specificity for *M. laxa* was evaluated under laboratory conditions for all samples. The results confirmed that this primer was specific for the *M. laxa* pathogen. However, ML86 SSR primer amplified smaller fragments (250 bp) in *M. fructigena*. This indicated that ML86 primer amplified fragments of different sizes in both species, allowing differentiation between them.

Under in silico conditions, all eight analyzed primers were specific either for *M. fructigena* or for *M. laxa* species. Analysis of the specificity of the eight primers under laboratory conditions using an m-PCR method showed that five primers were species-specific (Table 3).

The genetic polymorphism of *Monilinia* spp. was evaluated with ML2, ML86, MFg2, and MFg27 primers, marked with fluorescent FAM dye. Capillary electrophoresis confirmed the specificity of ML2 primer to the *M. laxa* pathogen (346 bp). In *M. fructigena* samples, the electrophoresis signal was too weak to be identified. ML2 primer was suitable for interspecific *Monilinia* spp. diversity analysis.

ML86 primer fragments were identified in both *Monilinia* spp. In *M. laxa*, alleles were homozygotic and heterozygotic in the 285–340 bp range. In *M. fructigena*, 11 homozygotic alleles were in the 250–309 bp range, and 23 heterozygotic alleles were in the 259–350 bp range. The PIC value of the primer was 0.83, showing high polymorphism and informativeness. The observed heterozygosity of the primer (Ho) was smaller (0.66) than the expected heterozygosity (0.84). MFg2 primer homozygotic fragments (263 bp) were identified only in the *M. fructigena* pathogen. MFg2 primer was suitable for interspecific *Monilinia* spp. genetic analysis. MFg27 primer did not show specificity for *Monilinia* spp., generating 244 bp alleles in both analyzed species.

The *Monilinia* spp. pathogens genetic marker database is available at www.lammc.lt, (http://193.219.178.20/ (accessed on 15 March 2024)). In the freely available online database, the information about SSR primers is available for three *Monilinia* species—*M. fructigena*, *M. laxa*, and *M. fructicola*. In the main window of the database, there is the option to choose the primer for one of three *Monilinia* species (Figure 4A). After selecting the pathogen and clicking on it, the primary information about the SSR primers is provided: marker names and forward and reverse sequences of the primers (Figure 4B). By clicking on the selected marker, additional information in the extended table (“Marker for SSR motive(s) info”) appears with the fragment size, SSR motif, repeat of the motif, and the name of the scaffold with the exact location with the coordinates (start and end) of the referenced sequenced scaffold (Figure 4C). In the database, there is a possibility to mark the markers as validated and/or unique. Any information uploaded on experimental validation of primers or information on species uniqueness is confirmed by the administrators. The information about the validation is shown in the additional info window (Figure 4D). The database is freely accessible worldwide, and all researchers working with these pathogens can continuously update the database with their experimental data.

## 4. Discussion

The increasing number of sequenced genomes with next-generation sequencing platforms improves the knowledge of genomics (genome structure, function, and dynamics) and transcriptomics of organisms [30]. Advancements in next-generation sequencing technologies and bioinformatics provide considerable data resources for microsatellite marker development. GMATA software allows for the analysis of large sequences with large capacity and fast processing [13]. For SSR mining and marker development, the GMATA tool has been used for various organisms: *Acacia pachyceras* Schwartz [31], spinach [32], brown trout (Salmo trutta) [33], *Nicotiana* [34], and olive (*Olea europaea* L.) [35]. However, the distribution of SSRs for *Monilinia* spp. has not been analyzed before due to the absence of whole genome sequences. This is the first report on SSR identification and distribution of whole *Monilinia fructigena*, *Monilinia laxa*, and *Monilinia fructicola* genomes.

In the *Monilinia* spp. pathogen genomes, dimer and trimer nucleotide types of microsatellite motifs were the most abundant. This structure of microsatellites was observed in other fungi, including *Colletotrichum falcatum* [36], *A. nidulans*, *C. neoformans*, *E. cuniculi*, *F. graminearum*, *M. grisea*, *N. crassa*, *S. cerevisiae*, *S. pombe*, and *U. maydis* [37]. The most frequent dinucleotide SSR motifs in the *Monilinia* spp. genomes were ‘TA’ and ‘AT’, observed in other fungi species, such as *A. nidulans*, *N. crassa*, *S. cerevisiae*, and *S. pombe* [37]. However, in *U. maydis*, *M. grisea*, and *Colletotrichum falcatum*, the most abundant dinucleotide SSRs were ‘TC’ and ‘CT’ [36,37]. In *M. fructicola* (19.7%), *M. fructigena* (24.6%), and *M. laxa* (26.2%), most of the microsatellites were short (10 bp), showing a low level of polymorphism. The observation of a higher number of microsatellites (SSRs) and increased SSR density in the *M. fructicola* pathogen, which possessed the largest genome among the analyzed pathogens, at 44,048 Mbp, suggested a positive correlation between genome size and microsatellite abundance [38]. In total, 9754 microsatellite markers were developed for *M. fructicola*, 8506 for *M. fructigena*, and 8106 for *M. laxa* pathogens, most of which (>9 5%) were species-specific under in silico conditions.

Under in silico conditions, ML2 primer size was 348 bp, while in analysis by capillary electrophoresis, the size of the generated fragments was 346 bp only in the *M. laxa* pathogen. MFg2 primer (in silico 260 bp) size under laboratory conditions was 263 bp in the *M. fructigena* pathogen. MFg27 (in silico 239 bp) size under laboratory conditions was 244 bp in the *M. laxa* and *M. fructigena* pathogens. However, due to the accuracy of capillary electrophoresis, ML86 primer (in silico 336 bp) size under laboratory conditions in the *M. laxa* pathogen was in the range of 285–340 bp and in *M. fructigena*, it was in the range of 250–350 bp. In silico-generated primer sizes corresponded to capillary electrophoresis results. In silico-generated primer pairs produced amplicons within the expected size range under laboratory conditions as in other studies [13]. The in silico PCR analysis indicated the specificity of all SSR markers to particular species. However, evaluation under laboratory conditions revealed that only five of the eight SSR primer pairs demonstrated specificity for the targeted species. This research shows the importance of validating primers developed under in silico conditions under laboratory conditions before using them in research.

The establishment of the initial microsatellite marker database for the genus *Monilinia*, featuring three distinct species—*Monilinia fructigena*, *Monilinia laxa*, and *Monilinia fructicola*, comprises a comprehensive repository of 26,366 markers. The usage of bioinformatic tools for the development of these simple sequence repeat markers enhances the efficiency of the process. The distribution of SSR markers across the species’ entire genome holds significant promise for exploring and understanding genomic variability [39]. The information in the database is helpful for genetic and molecular analysis of the pathogens and provides a basis for improving a database of single nucleotide polymorphism (SNP) information. Data from this research will be valuable in finding links between plants in the Rosaceae family and *Monilinia* spp. pathogenicity to develop resistance in *Monilinia* spp. plant genotypes.

## 5. Conclusions

This study analyzed the structure of SSR markers distributed in the genomes of the three most common *Monilinia* spp. pathogens worldwide, and an online freely accessible *Monilinia* spp. database was created. The database contains information about SSR markers of the three most common *Monilinia* spp. pathogens worldwide—*Monilinia fructigena*, *Monilinia laxa*, and *Monilinia fructicola*. The specificity of eight primer pairs was validated under laboratory conditions. The generated SSRs of *Monilinia* spp. expand molecular marker resources by providing data for future research and breeding. The organized molecular data allows researchers to identify, compare, and assess molecular data for specific needs. The *Monilinia* spp. marker database is publicly available at www.lammc.lt (http://193.219.178.20/ (accessed on 15 March 2024)).

## Figures and Tables

**Figure 1 microorganisms-12-00605-f001:**
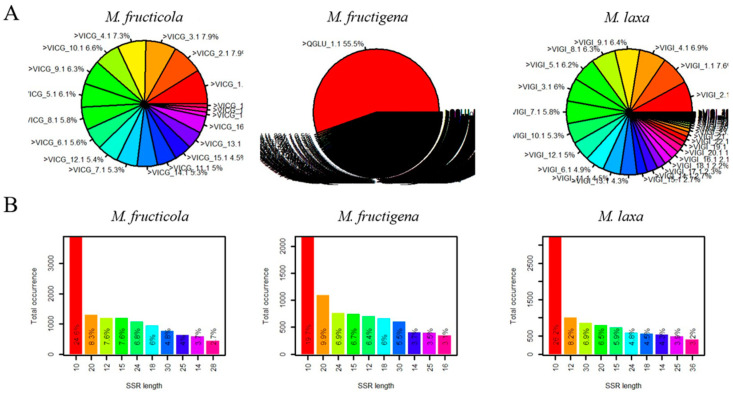
Distribution of SSR motifs in *Monilinia* spp. scaffolds (**A**) and their quantity according to the motif length (**B**).

**Figure 2 microorganisms-12-00605-f002:**
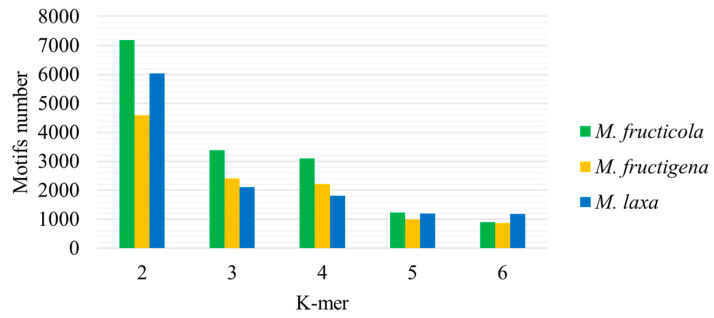
Microsatellite motif k-mer distribution in three *Monilinia* spp.

**Figure 3 microorganisms-12-00605-f003:**
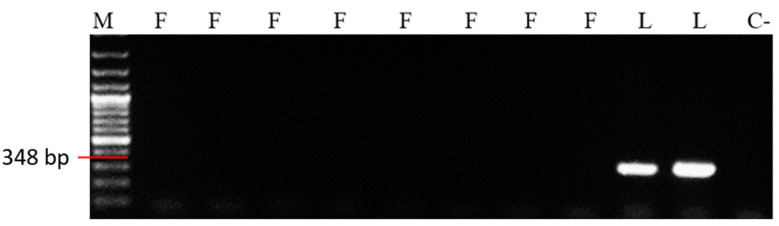
The specificity of ML2 primer for isolates. L: *M. laxa* sample; F: *M. fructigena* sample; M: 100 bp marker; C-: negative control.

**Figure 4 microorganisms-12-00605-f004:**
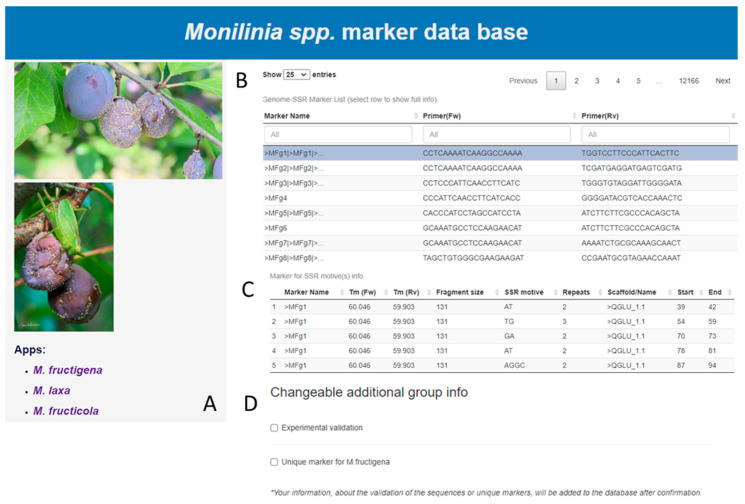
The *Monilinia* spp. genetic marker database. (**A**) Homepage of database, (**B**) general information about SSR primers, (**C**) extended table with additional information about the SSR primers, (**D**) information about primer validation and uniqueness.

**Table 1 microorganisms-12-00605-t001:** Primers specific for *Monilinia* species (MFg—*M. fructigena*, ML—*M. laxa*) developed for SSR analysis and their characterization.

Marker Name	Sequence		T (°C)
MFg2 *	F	ACTCTCGTCTCCACCTTCCA	60
	R	CCTGAAGGATAGCACCCTGA	60
MFg27 *	F	CCTTCAAATGGGCAAGATGT	56
	R	TGAATGTTGGTGAGGCGTTA	56
MFg39	F	GGTTTCTGCCAAAAGTCTCC	58
	R	GTAGGTGATGGCGCTGTTTT	58
MFg90	F	ACCGATTCCAGTTGATGGAG	58
	R	ATCGGTCCATGATTGTCGTT	56
ML2 *	F	TCGTGAACTTTACTCTCGTCTCC	63
	R	ATGTCGTTCCAGAAGGCACT	58
ML86 *	F	CAAGGACGTTTCCAAAGCAT	56
	R	ACATCTTGCCCATTTGAAGG	56
ML104	F	AAGTCTCCTCCTCGCAGCTT	60
	R	TCCGTTGGGCTTGTAGTTTC	58
ML159 *	F	ATTCATGCTCAGCGAACCTT	56
	R	GATCTCACGCCTCCAGCTAC	63

* Experimentally validated as *Monilinia* species-specific primers.

**Table 2 microorganisms-12-00605-t002:** SSR distribution in *Monilinia* spp. genomes.

Species	No. of SSR Motifs	Genome Size (Mbp)	SSR Density (per Mbp)	Developed SSR Markers	Species-Specific SSR Markers
*M. fructicola*	15,788	44,048	359	9754	9617 (98.6%)
*M. fructigena*	11,091	39,329	284	8506	8188 (96.3%)
*M. laxa*	12,337	42,815	286	8106	7781 (96.0%)
Total	39,216	-	-	26,366	25,586

**Table 3 microorganisms-12-00605-t003:** Pathogenesis-related SSR primer specificity for *Monilinia* spp.

SSR Primer	Fragment Size (bp)	In Silico Specificity	Specificity under Laboratory Conditions
MFg27	239 bp	*M. fructigena*	*M. fructigena*
MFg39	346 bp	*M. fructigena*	-
MFg90	292 bp	*M. fructigena*	-
ML2	348 bp	*M. laxa*	*M. laxa*
ML86	336 bp	*M. laxa*	*M. laxa*
ML104	358 bp	*M. laxa*	-
ML159	313 bp	*M. laxa*	*M. laxa*
MFg2	260 bp	*M. fructigena*	*M. fructigena*

## Data Availability

Data are contained within the article.

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
