# Peer review of "One Step Forwards in Knowledge of Blossom Blight Brown Rot Disease: Monilinia spp. SSR Marker Database"

_microorganisms, 2024, doi:10.3390/microorganisms12030605_

Round 1

Reviewer 1 Report

Comments and Suggestions for Authors

This article provides valuable information about the creation of a freely available Monilinia spp. marker database, which contains microsatellites (SSR) data of three significant fungal pathogens: M. fructigena, M. laxa, and M. fructicola.  Overall, the language used in the article is clear and concise, but there are some minor grammatical errors and awkward phrasings that could be corrected to improve readability. I also suggest including information about any quality control measures implemented during the experiment, such as the use of positive and negative controls or replicates. Additionally, mentioning any steps taken to ensure the accuracy and reproducibility of the capillary electrophoresis results would strengthen the experimental procedure.

Were the observed DNA fragments consistent with the expected sizes based on the SSR markers and primer sequences? Quality of figure 3 and 4 need improvement. The discussion section of the article effectively summarizes the findings of the study and provides valuable insights into the distribution and characteristics of SSR markers in Monilinia spp. genomes

Authors indicated that “This is the first report of SSR identification and 269 distribution of Monilinia fructigenaMonilinia laxa and Monilinia fructicola genomes”, however, I found similar related studies such as https://apsjournals.apsnet.org/doi/full/10.1094/PHYTO-03-18-0074-R therefore I recomend to discuss these studies 

Comments on the Quality of English Language

Overall, the language used in the article is clear and concise, but there are some minor grammatical errors and awkward phrasings that could be corrected to improve readability.

Reviewer 2 Report

Comments and Suggestions for Authors

Monilinia spp. fungi are serious and widespread pathogens of fruit trees, causing brown rot flower blight. To control Monilinia spp., it is crucial to track outbreaks and genetic changes in these pathogens. Microsatellite markers are usually used for this purpose because they are highly polymorphic, easy to use and cheap. The authors used a modern approach based on in silico analysis of the sequenced genomes of M. fructigena, M. laxa and M. fructicola. Based on the data obtained, the authors developed a database currently containing information on several dozen thousand SSR motifs and made it publicly available. Compared to the previous state of knowledge, when just over 20 SSR markers were known, the authors managed to achieve great progress. The manuscript is scientifically sound and very interesting. Ethical issues do not raise concern.

Minor remarks:

Lines 73 -74. Genome-wide Microsatellite Analysing Toward Application (GMATA) software [13].

In the case of software, not only publication is necessary, but also information about where the software is available.

Line 145 "using the PowerMaker program [25]"

The same, link provided in the Liu and Muse does not exist.

Figure 3. The specificity of ML2 primer for isolates. In picture – L– M. Laxa sample.

Change "In picture – L– M" now it looks like at figure are present samples from L to M.

Reviewer 3 Report

Comments and Suggestions for Authors

The manuscript documents the development of a web server which contains simple sequence repeats (SSRs) from three fungal pathogens, M. Fructigena; M. laxa; and M. fructicola. In total, more than 39,000 SSR motifs and 26,000 markers are collected and stored in the database. The effort is definitely worthwhile and would be appreciated by the community as the web server provides a platform on which researchers could share their findings and take advantage of the comprehensive database to perform in silico validation using PCR. Hopefully, SSR markers of M. polystroma will also be included in the future.
